

# Oxygen and nutrient trends in the Tropical Oceans

Lothar Stramma and Sunke Schmidtko

GEOMAR Helmholtz Centre for Ocean Research Kiel, Düsternbrooker Weg 20, 24105 Kiel, Germany

*Correspondence to*: Lothar Stramma (lstramma@geomar.de)

**Abstract.** A vertical expansion of the intermediate-depth low-oxygen zones (300 to 700 m) is seen in time series for selected tropical areas for the period 1960 to 2008, in the eastern tropical Atlantic, the equatorial Pacific and the eastern tropical Indian Ocean. These nearly five decade-long time series were extended to 68 years by including rare historic data starting in 1950 and more recent data. For the extended time series between 1950 and 2018 the deoxygenation trend for the layer 300 to 700 m is similar to the deoxygenation trend seen in the shorter time series. Additionally, temperature, salinity and nutrient time series in the upper ocean layer (50 to 300 m) of these areas were investigated since this layer provides critical pelagic habitat for biological communities. Generally, oxygen is decreasing in the 50 to 300 m layer except for an area in the eastern tropical South Atlantic. Nutrients also showed long-term trends in the 50 to 300 m layer in all ocean basins and indicates overlying variability related to climate modes. Nitrate increased in all areas. Phosphate also increased in the Atlantic and Indian Ocean areas, while it decreased in the two areas of the equatorial Pacific Ocean. Silicate decreased in the Atlantic and Pacific areas but increased in the eastern Indian Ocean. Hence oxygen and nutrients show trends in the tropical oceans, though nutrients trends are more variable between ocean areas than the oxygen trends, therefore we conclude that those trends are more dependent on local drivers in addition to a global trend. Different positive and negative trends in temperature, salinity, oxygen and nutrients indicate that oxygen and nutrient trends cannot be completely explained by local warming.

## 1 Introduction

Temperature, oxygen and nutrient changes in the ocean have various impacts on the ecosystem. These impacts span from habitat compression in the open ocean (Stramma et. al., 2012) and affect all marine organisms through multiple direct and indirect mechanisms (Gilly et al., 2013) to affect the ecophysiology of marine water-breathing organisms with regard to distribution, phenology and productivity (Cheung et al., 2013). Despite its far-reaching consequences for humanity, the focus on climate change impacts on the ocean lags behind the concern for impacts on the atmosphere and land (Allison and Bassett, 2015). An oceanic increase in stratification, thus reduction in ventilation as well as decrease of oceanic dissolved oxygen are two of the less obvious but important expected indirect consequences of climate change on the ocean (Shepherd et al., 2017). Warming leads to lighter water in the surface layer and increased stratification reducing the mixing and deep ventilation of oxygen-rich surface water to the subsurface layers. Increasing ocean stratification over the last half century of



about 5% is observed in the upper 200 m (Li et al. 2020). The subsequent previously observed deoxygenation (e.g. Stramma
et al, 2008, Schmidtko et al 2017) of the open ocean is one of the major manifestations of global change. This temperature
oxygen relation can also be seen for the 0-1000 m layer of the global ocean, as the oxygen inventory is negatively correlated
with the ocean heat content (r=-0.86; 0-1000 m) (Ito et al., 2017). Oxygen-poor waters often referred to as oxygen minimum
zones (OMZ) occupy large volumes of the intermediate-depth eastern tropical oceans. In an investigation of six selected
areas for the 300 to 700 m layer in the tropical oceans for the time period 1960 to 2008 Stramma et al. (2008) observed
declining oxygen concentrations of -0.09 to -0.34 µmol kg$^{-1}$ year$^{-1}$ and a vertical expansion of the intermediate depth low
oxygen zone. Such a vertical expansion of the OMZ that is entered and passed by diel vertical migrators and sinking
particles could have widespread effects on species distribution, the biological pump and benthic-pelagic coupling (Wishner
et al., 2013).  The areas of the world ocean investigated for oxygen changes can be extended and in a quantitative assessment
of the entire world ocean oxygen inventory by analysing dissolved oxygen and supporting data for the complete oceanic
water column over the past 50 years since 1960. Schmidtko et al. (2017) reported that the global oceanic oxygen content of
227.4 ± 1.1 petamoles (10$^{15}$mol) has decreased by more than two percent (4.8 ± 2.1 petamoles). However, these oxygen
changes vary by region with some areas showing increasing oxygen values on time scales related to climate modes.
The nutrient distribution is in addition to oxygen a key parameter controlling the marine ecosystems. However, very little is
known about long term nutrient changes in the ocean. The transformation of carbon and nutrients into organic carbon, its
sinking into the in the deep ocean, and its decomposition at depth, is known as the biological carbon pump. As a
consequence, nutrients are consumed and thus lower in the surface ocean and released and thus higher in the deep ocean. The
oceanic distribution of nutrients and patterns of biological production are controlled by the interplay of biogeochemical and
physical processes, and external sources (Williams and Follows, 2003). In the upper 500 to 1000 m of the tropical oceans the
nutrient concentration is higher than in the subtropics and is decreasing westwards (Levitus et al., 1993). In the subarctic
North Pacific surface nutrient concentration decreased during 1975 to 2005, and is strongly correlated with a multidecadal
increasing trend of sea surface temperature (SST) (Ono et al., 2008). Below the surface, however, oxygen decreased and
nutrients increased in the subarctic Pacific pycnocline from the mid-1980s to around 2010 (Whitney et al., 2013). Nutrients
would be expected to vary inversely with oxygen, if the dominant process was the remineralization of marine detritus
(Whitney et al., 2013). In a recent study the trends of nutrients in the open Pacific Ocean were investigated (Stramma et al.,
2020) and in the open Pacific Ocean nutrient trends were observed and seemed to be related to oxygen trends. The supply of
nutrients to the sunlit surface layer of the ocean has traditionally been attributed solely to vertical processes. However,
horizontal advection may also be important in establishing the availability of nutrients in some regions. Palter et al. (2005)
showed that the production and advection of North Atlantic Subtropical Mode Water introduces spatial and temporal
variability in the subsurface nutrient reservoir beneath the North Atlantic subtropical gyre. By means of a coupled ecosystem
circulation model Oschlies (2001) described for the North Atlantic that the long term change in the North Atlantic
Oscillation (NAO; e.g. Hurell and Deser, 2010) between the 1960s and 1990s may have induced significant regional changes
in the upper ocean's nutrient supply. These include a decrease of nitrate supply to the surface waters of by about 30% near





Bermuda and in mid latitudes, and a simultaneous 60% increased nitrate flux in the upwelling region off West Africa. On the
other side of the globe the Indonesian throughflow (ITF) is a chokepoint in the upper ocean thermohaline circulation,
carrying Pacific waters through the strongly mixed Indonesian Seas and into the Indian Ocean (Ayers et al., 2014). Ayers et
al. (2014) determined the depth- and time-resolved nitrate, phosphate, and silicate fluxes at the three main exit passages of
the ITF: Lombok Strait, Ombai Strait, and Timor Passage. Nutrient flux as well as its variability with depth and time differed
greatly between the passages. They estimated the effective flux of nutrients into the Indian Ocean and found that the majority
of ITF nutrient supply to the Indian Ocean is to thermocline waters, where it is likely to support new production and
significantly impact Indian Ocean biogeochemical cycling.
Here we investigate the extend of changes in oxygen, temperature and salinity trends for the six tropical areas with longer
time series compared to the previously about one third shorter timeseries.   Additionally, trends in the biological active near
surface layer 50 to 300 m are investigated. As the upper ocean provides critical pelagic habitat for biological communities,
nutrient time series of the six tropical areas since 1950 are investigated at 50 to 300 m depth, as nutrient changes in
combination with hydrographic changes will influence the biological productivity of the ocean (Sigman and Hain, 2012).
The upper boundary of 50 m was chosen to reduce the influence of the seasonal cycle in the upper 50 m although the
seasonal cycle in the tropics is weaker than in most subtropical and subpolar regions (Louanchi and Najjar, 2000). As there
are indications that climate modes and the El Niño-Southern Oscillation (ENSO) events have an influence on the trends, we
check whether these signals are apparent in the data in the near surface layer.
**2 Data and methods**
Stramma et al. (2008) investigated the temperature and oxygen trends for the period 1960 to 2008 in the 300 to 700 m layer
of six tropical ocean areas. There were three areas in the tropical Atlantic (A: 10°–14°N, 20°–30°W; B: 3°S–3°N, 18°–
28°W; C: 14°S–8°S, 4°–12°E), two areas in the eastern and central tropical Pacific (D: 5°S–5°N, 105–115°W; E: 5°S–5°N,
165°–175°W) and one in the eastern Indian Ocean (F: 5°S–0°N, 90°–98°E) (Figure 1). Here these time series were extended
with more recent data as well as back in time to 1950 for the regions with available data (Table 1 and Figure 2).



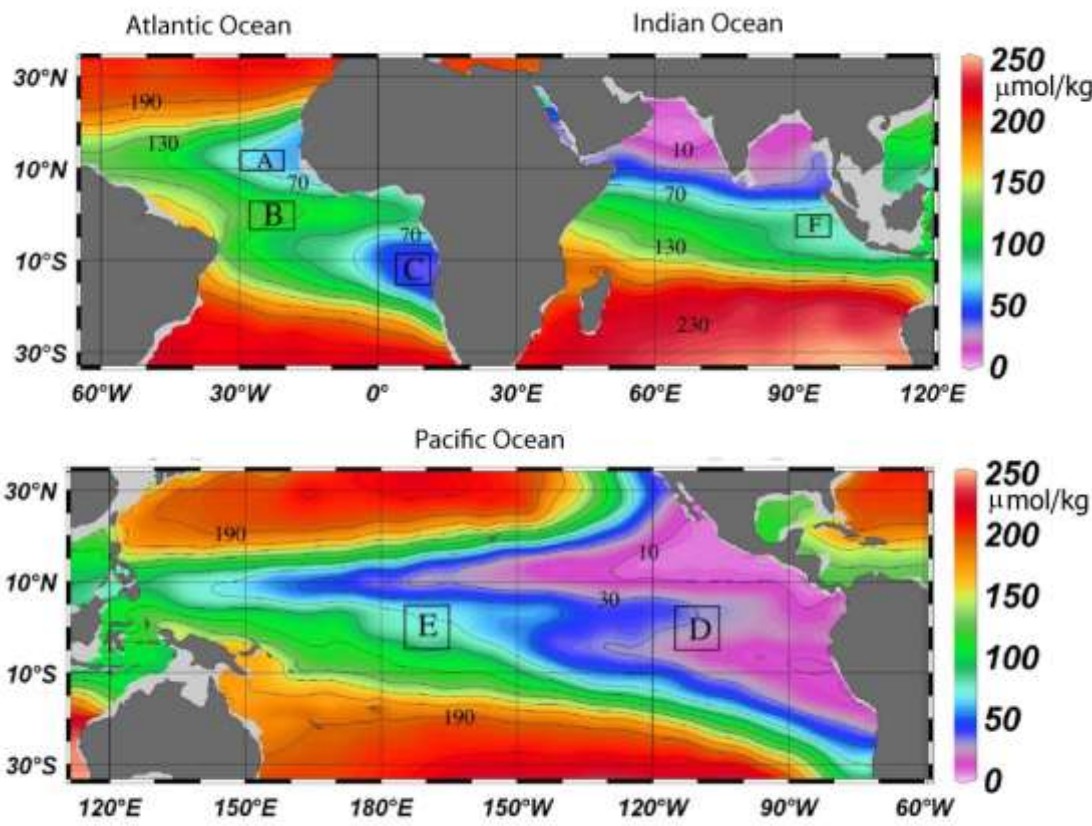

**Figure 1:** Climatological mean dissolved oxygen concentration (µmol kg$^{-1}$ shown in color) at 400 m depth contoured at 20 µmol kg$^{-1}$ intervals from 10 to 230 µmol kg$^{-1}$ (black lines). Analysed areas A to F (Table 1) are enclosed by black boxes (Stramma et al., 2008).

Despite long-term trends in ocean oxygen also climate signal related influence on the trends was observed in recent years. More recently also long-term trends and climate signal related influence was observed for nutrients. The areas D and E were also used for the layer 50 to 300 m for oxygen changes in Stramma et al. (2020), but not for nutrient trends due to the low amount of available nutrients data. However, here we list also the nutrients trends for these two areas, despite the low amount of data does not make these calculations statistically significant (Table 2).

The main hydrographic data set is similar to the one used and described in Schmidtko et al. (2017), relying on Hydrobase and World Ocean Database bottle data for nutrient data. Quality control and handling is described in Schmidtko et al. (2017) for oxygen and is used here similarly for nutrients. The only divergence to the described procedure was that bottle data with missing temperature and/or salinity were assigned the temporal and spatial interpolated temperature and salinity derived from MIMOC (Schmidtko et al., 2013). This was done to ensure all data were in µmol kg$^{-1}$ and not requiring the discarding





of already sparse data due to missing water density values. This enables us to use data in mol l$^{-1}$ or ml l$^{-1}$ which otherwise
could not be used.
In the Atlantic the hydrographic and nutrient data were extended with some *RV Meteor*, *RV Merian* and *RV Poseidon*
cruises. For the area A data from Meteor cruises M68/2 (2006), M83/1 (2008), M97 (2010), M119 (2015) and M145 (2018)
and Merian MSM10/1 (2008) were added. For area B Meteor cruises M106 (2014), M130 (2016) and M145 (2018) were
added. For area C cruise data from Poseidon P250 (1999), Merian MSM07 (2008), Meteor M120 (2015), Meteor M131
(2016) and Meteor M148 (2018) were included.
The Pacific the region at 5°N–5°S, 165–175°W (area E) which had data until 2009 was supplemented with data from a *RV*
*Investigator* cruise at 170°W from June 2016. The region 5°N–5°S, 105–110°W (area D), which had data up to 2008, was
supplemented with data from a *RV Ron Brown* cruise at 110°W in December 2016.
Climate indices considered include the NAO, the AMO, the PDO, ENSO, as well as the Indian Ocean Dipole Mode (IOD).
The NAO is an extratropical climate signal of the North Atlantic. As our areas are tropical regions the three Atlantic areas
were investigated relative to the Atlantic Multidecadal Oscillation (AMO) index (Montes et al., 2016) before and after 1995.
The AMO was high before 1963, low until 1995 and high since 1995.  In the Pacific the central equatorial area at 5°N-5°S,
165°-175°W (area E in Stramma et al., 2008) which had hydrodata until 2009, was supplemented with data from a *R/V*
*Investigator* cruise at 170°W from June 2016. The eastern equatorial area 5°N-5°S, 105°-115°W (area D in Stramma et al.
2008), which had hydrodata until 2008, was supplemented with data from a RV *Ron Brown* cruise at 110°W in December
2016. The data were investigated in relation to the Pacific Decadal Oscillation (PDO; e.g. Deser et al., 2010) before and after
1977. The PDO was negative from 1944 to 1976, positive from 1977 to 1998, variable from 1998 to 2013 and positive after
2013. In the Indian Ocean the available data covered the area F only after 1960 but until 2016. The area F (0° to 5°S, 90° to
98°E) is shown in relation to the IOD (Saji et al., 1999), which slightly increased after 1990.
Linear trends and their 95% confidence interval were computed using annual averages (all measurements from one year were
attributed to that year) of the profiles linearly to standard vertical depth levels. A computation routine was used which used
the effective number of degrees of freedom for the computation of the confidence interval. The data used for the oxygen time
series were interpolated with an objective mapping scheme (Bretherton et al., 1976) with Gaussian weighting. In the 50 to
300 m layer and the 300 to 700 m a temporal half folding range of 0.5 year and a vertical half folding range of 50 m with
maximum ranges of 1 year and 100 m were applied. The covariance matrix was computed from 100 local data points and 50
random data points within the maximum range, for the diagonal of the covariance matrix a signal to noise ratio of 0.7 was set
(see Schmidtko et al. 2013, for details). A more improved mapping scheme was used compared to the one used in Stramma
et al. (2008) where larger temporal ranges were used (1-year half folding and a maximum temporal range of 2 years).
Nutrients nitrite ($NO_2^-$), nitrate ($NO_3^-$), phosphate ($PO_4^{3-}$) and silicic acid ($Si(OH)_4$ referred to as silicate hereafter) on the
recent cruises were measured on-board with a QuAAtro auto-analyzer (Seal Analytical). For recent autoanalyzer
measurements precisions are 0.01 µmol kg$^{-1}$ for phosphate, 0.1 µmol kg$^{-1}$ for nitrate, and 0.5 µmol kg$^{-1}$ for silicate and 0.02
mL L$^{-1}$ (~ 0.9 µmol kg$^{-1}$) for oxygen from Winkler titration (Bograd et al., 2015). For older uncorrected nutrient data, offsets





are estimated to be 3.5% for nitrate, 6.2% for silicate and 5.1% for phosphate (Tanhua et al., 2010). One problem with
nutrient data is that certified reference material (CRM) was applied to some measurements while for other measurements
only a bias was applied.
The ENSO cycle of alternating warm El Niño and cold La Niña events is the climate system's dominant year-to year signal.
ENSO originates in the tropical Pacific through interaction between the ocean and the atmosphere, but its environmental and
socioeconomic impacts are felt worldwide (McPhaden et al., 2006). Three month running mean SST anomalies (ERSST.v5
SST anomalies) in the Niño 3.4 region (equatorial Pacific: 5°N to 5°S, 120°W to 170°W) of at least +0.5°C and lasting for at
least 5 consecutive three months periods are defined as El Niño events and 5 consecutive three months periods of at least -
0.5°C are defined as La Niña events (http://origin.cpc.ncep.noaa.gov/products/analysis_monitoring/ensostuff/ONI_v5.php).
In case of measurements in ENSO years in figures 3, 4 and 5 the very strong El Niño events of 1983, 1998 and 2015 and the
strong El Niño events 1957, 1965, 1972, 1987 and 1991 are marked by red circles and the strong La Niña events 1974, 1976,
1989, 1999, 2000, 2007 and 2010 are marked by blue squares in these years. A shoaling thermocline, such as occurs in the
eastern Pacific during La Niña or cool (negative) PDO state, enhances nutrient supply and organic matter export in the
eastern Pacific while simultaneously increasing the fraction of that organic matter that is respired in the low-oxygen water of
the uplifted thermocline. The opposite occurs during El Niño or a warm (positive) PDO state; a deeper thermocline reduces
both export and respiration in low-oxygen water in the eastern Pacific, allowing the hypoxic water volume to shrink
(Deutsch et al., 2011; Fig. S7). ENSO also has some influence on the tropical Atlantic and Indian Oceans. The equatorial
Atlantic oscillation is influenced by the Pacific ENSO with the equatorial Atlantic sea surface temperature lagging by about
six months (Latif and Grötzner, 2000). In the Indian Ocean a recent weakening of the coupling between the ENSO and the
IOD mode after the 2000s and 2010s compared to the previous two decades (1980s and 1990s) (Ham et al., 2017).


**3 Trends in temperature, salinity, oxygen and nutrients**
**3.1 Trends in the 300 to 700 m depth layer**
Nutrient data are sparse in the deeper part of the ocean and are less important than the near surface layer for the marine
ecosystems and therefore are not presented here for the 300 to 700 m depth layer. Oxygen trends for the period 1960 to 2008
for the 300 to 700 m layer of the six areas investigated (Stramma et al., 2008) for the tropical oceans were all negative in the
range -0.09 to -0.34 µmol kg$^{-1}$ year$^{-1}$ (Table 1). For the extended time period between 1950 and 2018 the oxygen trends were
in the same order of magnitude for the areas A to F in the range -0.11 to -0.27 µmol kg$^{-1}$ year$^{-1}$ (Table 1).  The 1950 to 2018
temperature trends were positive in the three Atlantic areas and the eastern tropical Pacific, but negative in the central Pacific
and Indian Ocean areas (Table 1). In the eastern tropical Pacific (area D) and the eastern Indian Ocean (area F) there was
even a reversed trend in temperature compared to the shorter time period between 1960 and 2008, although all temperature

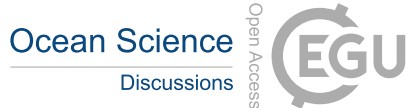

trends are not within the 95% confidence interval difference from 0. The salinity of the 300 to 700 m layer increased for the
Atlantic and Indian Ocean areas and decreased in the two Pacific areas (Table 1).

**Table 1.**   Linear trends (300 to 700 m) of temperature in °C yr$^{-1}$, oxygen in µmol kg$^{-1}$ yr$^{-1}$ and salinity yr$^{-1}$ with 95%
confidence intervals (p-values) where data are available for the entire period listed.  Trends whose 95% confidence interval
includes zero are shown in *italics*. Trends computed in Stramma et al. (2008) are shown for comparison.

| Parameter | trend   time period   depth layer | (Stramma et al., 2008) |
|---|---|---|
| Area A | 10°N-14°N, 20°W-30°W | |
| Temperature | +0.009 ± 0.005 1952-2018  300-700 m | +0.009 ± 0.008 1960-2006  300-700 m |
| Oxygen | -0.27 ± 0.12 1952-2018   300-700 m | -0.34 ± 0.13 1960-2006  300-700 m |
| Salinity | +0.0012 ±0.0009 1952-2018 300-700 m | |
| | | |
| Area B | 3°S-3°N, 18°W-28°W | |
| Temperature | +0.005 ± 0.004 1952-2018  300-700 m | *+0.005 ± 0.008* 1960-2006  300-700 m |
| Oxygen | *-0.25 ± 0.65* 1952-2018   300-700 m | -0.19 ± 0.12 1960-2006  300-700 m |
| Salinity | *+0.0001 ± 0.0005* 1952-2018 300-700 m | |
| | | |
| Area C | 14°S-8°S, 4°E-12°E | |
| Temperature | +0.006 ± 0.004 1950-2018   300-700 m | *+0.002 ± 0.011* 1961-2008  300-700 m |
| Oxygen | -0.11 ±  0.100 1950-2018 300-700 m | -0.17 ± 0.11 1961-2008  300-700 m |
| Salinity | *+0.0005 ±0.0009* 1950-2018 300-700 m | |
| | | |
| Area D | 5°S-5°N, 105°W-115°W | |
| Temperature | *+0.003 ±0.004* 1955-2016  300-700 m | *-0.001 ± 0.009* 1962-2006  300-700 m |
| Oxygen | -0.24 ± 0.15 1957-2016   300-700 m | *-0.13 ± 0.32* 1962-2006  300-700 m |
| Salinity | *-0.0001 ±0.009* 1950-2016 300 -700 m | |
| | | |
| Area E | 5°S-5°N, 165°W-175°W | |
| Temperature | *-0.001 ±0.011* 1950-2016  300-700 m | -0.010 ± 0.008 1961-2006  300-700 m |
| Oxygen | *-0.18 ±0.25* 1950-2016  300-700 m | *-0.19 ± 0.20* 1961-2006  300-700 m |
| Salinity | *-0.0003 ±0.0009* 1950-2016 300-700 m | |
| | | |
| Area F | 5°S-0°N, 90°E-98°E | |





| 203 | Temperature | *-0.004 ± 0.010* 1960-2016  300-700 m | *+0.005 ± 0.007* 1960-2007  300-700 m |
| 204 | Oxygen | *-0.13 ± 0.17* 1960-2016  300-700 m | *-0.09 ± 0.21* 1960-2007  300-700 m |
| 205 | Salinity | *+0.0001 ± 0.0010* 1960-2016 300-700 m | |

For the area A (10°-14°N, 20°-30°W) the oxygen trend for 300 to 700 m for the period 1952 to 2018 (Figure 2a) was weaker
(-0.27 ± 0.12 µmol kg$^{-1}$ yr$^{-1}$) than for the period 1960 to 2006 (-0.34 ± 0.13 µmol kg$^{-1}$ yr$^{-1}$). In the western subtropical and
tropical Atlantic oxygen measurements from time series stations as well as shipboard measurements showed a significant
relationship with the wintertime AMO index (Montes et al., 2016). During negative wintertime AMO years trade winds are
typically stronger and these conditions stimulate the formation and ventilation of Subtropical Underwater (Montes et al.,
2016) with higher oxygen content. Even in the 300 to 700 m layer of Area A (Figure 2a) as well as the 50 to 300 m layer
(Figure 3a) the oxygen content is higher during the negative AMO period and lower during the positive AMO phase. For a
section along 23°W between 6°–14°N from 2006 to 2015 crossing area A an oxygen decrease in the 200 to 400 m layer and
an increase in the 400 to 1000 m layer was described (Hahn et al., 2017) which can't be confirmed in area A due to the
different geographical and temporal boundaries and the variable annual mean oxygen values after 2006 in area A.
The 1952 to 2018 oxygen trend in the equatorial Atlantic (area B) shows a large 95% confidence interval, different to the
shorter time period 1960 to 2006 (Table 1). The larger confidence interval is caused by a low oxygen concentration in 1952
and large variability after 2006 (Figure 2b). The equatorial Atlantic in the depth range 500 to 2000 m is influenced by
Equatorial Deep Jets with periodically reversing flow direction influencing the transport of oxygen (Bastin et al., 2020)
which might be one reason of the large oxygen variability. During the negative AMO the oxygen trend was slightly positive
(-0.034± 1.39 µmol kg$^{-1}$ yr$^{-1}$) but negative after 1995 (Figure 2b).

**Figure 2:** Annual mean oxygen concentration for years available and trends for the layer 300 to 700 m in µmol kg⁻¹ plotted for the available years in the time period 1950 to 2018 (dashed red line) and for the positive and negative periods of the AMO in the Atlantic (a-c), the PDO in the Pacific (d,e) and the IOD in the Indian Ocean (f) as solid red lines. The AMO, PDO and IOD are shown as grey lines. The change of AMO status in 1963 and 1995, the change of the PDO phase in 1977, 1999 and 2013 and the IOD in 1990 are marked by dotted vertical lines.

The area C in the eastern tropical South Atlantic shows similar positive trends in temperature and salinity (Table 1) as in the two other Atlantic areas investigated. Area C in located in the region with the lowest oxygen content in the Atlantic Ocean





(Figure 1). Due to the already low oxygen concentration in this region the decrease in oxygen is weaker than in the two other
Atlantic Ocean areas in the period 1950 to 2018, similar to the weaker decrease in area C for the shorter time period 1961 to
2008 (Table 1). Higher oxygen concentrations were also seen in the few oxygen profiles in area C during the negative AMO
and lower oxygen concentrations were measured after the year 2000 (Figure 2c).
In the equatorial Pacific the two areas show a clear long-term oxygen decrease in the 300 to 700 m layer, but no clear
changes related to the PDO phases before and after 1977 (Figure 2d,e). However, the PDO-index after 1977 was mainly
positive until 1999 and mainly negative between 1999 and 2013. In case these time periods are looked at separately the
oxygen concentration was higher during the period 1977 to 1990 and lower during 1999 to 2010 as expected for the PDO
influence (e.g. Deutsch et al. 2011).
In the eastern Indian Ocean, the 300 to 700 m oxygen concentration was lower for the slightly positive IOD phase after 1990
leading to a long-term oxygen concentration decrease in area F although the trends for the shorter periods prior to 1990 and
after 1990 showed a positive oxygen trend (Figure 2f), which are caused by high oxygen concentrations near the end of both
measurement periods. The temperature in this area decreased and salinity showed barely any change (Table 1), hence the
oxygen decrease is not coupled to temperature or hydrographic water mass changes.

**3.2 Trends in the 50 to 300 m layer**
The trend computations for the layer 50 to 300 m for temperature, salinity, oxygen and nutrients (Table 2) show different
trends for the selected areas in the three tropical oceans. In the near surface layer 50 to 300 m the long-term oxygen trends
were negative as in the deeper layer 300 to 700 m, except for area C in the eastern tropical South Atlantic (Figure 3c).
However, this oxygen trend in area C is not stable due to the large variability in the time period 1960 to 1990. The upper
layer of the area C is influenced by the Angola Dome centered at 10°S, 9°E (Mazeika, 1967) which might influence the
larger variability near the surface. The area C shows the largest mean nitrate, silicate and phosphate concentrations in the
Atlantic in the 50 to 300 m layer as well as the 300 to 700 m layer (Table 3) and shows the large nutrient availability in the
eastern tropical South Atlantic. At 250 m and 500 m depth the region of area C was shown with the highest nitrate and
phosphate concentrations of the tropical and subtropical Atlantic Ocean (Levitus et al. 1993). It was observed that in the
Pacific Ocean nutrient are related to oxygen changes and climate variability (Stramma et al., 2020). The ENSO signal was
apparent in most cases as in the tropical Atlantic and Indian Ocean (Nicholson, 1997) hence the oxygen distribution for the
layer 50 to 300 m (Figure 3) is marked for El Niño and La Niña events to check for the possible influence of ENSO in the
shallow depth layer. Most of the nutrient trends are due to sparse data coverage not statistically significant, nevertheless it is
insightful to compare the nutrient trends with the oxygen trends as well as the climate signals.



**Table 2.** Linear trends (50-300 m) of temperature in °C yr$^{-1}$, salinity yr$^{-1}$ and solutes in µmol kg$^{-1}$ yr$^{-1}$ with 95% confidence intervals (p-values) where data are available for the entire period 1950 to 2018 (left rows) and for the earlier time period (center rows) and later time period (right rows) separated in 1995 in the Atlantic Ocean (areas A, B, C), in 1977 in the Pacific Ocean (areas D, E) and 1990 in the Indian Ocean (area F).. Trends whose 95% confidence interval includes zero are shown in *italics*.

| Parameter | trend | time period | trend | time period | trend | time period |
|---|---|---|---|---|---|---|
| Area A | 10°N-14°N, 20°W-30°W, 50 -300 m | | | | | |
| Temperature | *+0.007 ± 0.008* | *1952-2018* | *+0.004 ± 0.021* | *1952-1993* | *-0.001 ± 0.050* | *2001-2018* |
| Salinity | *+0.0009 ± 0.0012* | *1952-2018* | *+00.27 ± 0.0033* | *1952-1993* | *+0.006 ± 0.0083* | *2001-2018* |
| Oxygen | -0.329 ± 0.231 | 1952-2018 | *-0.387 ± 0.639* | *1952-1993* | *+0.131 ± 1.120* | *2001-2018* |
| Nitrate | *+0.038 ± 0.077* | *1952-2018* | +0.112 ± 0.116 | 1952-1993 | *-0.022 ± 0.581* | *2001-2018* |
| Silicate | -0.066 ± 0.086 | 1952-2018 | *+0.002 ± 0.310* | *1952-1989* | *+0.029 ± 0.151* | *2001-2018* |
| Phosphate | *+0.001 ± 0.004* | *1952-2018* | *-0.002 ± 0.010* | *1952-1993* | *-0.024 ± 0.029* | *2001-2018* |
| | | | | | | |
| Area B | 3°S-3°N, 18°W-28°W, 50-300 m | | | | | |
| Temperature | *-0.007 ± 0.012* | *1952-2018* | *-0.013 ± 0.028* | *1952-1995* | *-0.017 ± 0.042* | *1997-2018* |
| Salinity | *+0.0003 ± 0.0011* | *1952-2018* | *+0.0001 ± 0.0030* | *1952-1994* | *+0.0010 ± 0.0040* | *1997-2018* |
| Oxygen | *-0.172 ± 0.421* | *1952-2018* | *-0.174 ± 0.874* | *1952-1994* | *-1.050 ± 2.010* | *1999-2018* |
| Nitrate | *+0.022 ± 0.075* | *1961-2018* | *+0.095 ± 0.111* | *1961-1994* | *+0.055 ± 0.369* | *1997-2018* |
| Silicate | -0.061 ± 0.041 | 1961-2018 | *-0.079 ± 0.107* | *1961-1994* | *-0.056 ± 0.144* | *1999-2018* |
| Phosphate | *+0.001 ± 0.004* | *1952-2018* | +0.007 ± 0.005 | 1952-1994 | *+0.003 ± 0.021* | *1997-2018* |
| | | | | | | |
| Area C | 14°S-8°S, 4°E-12°E, 50-300 m | | | | | |
| Temperature | *+0.006 ± 0.024* | *1950-2018* | +0.018 ± 0.020 | 1950-1994 | *+0.04 ± 0.108* | *1995-2018* |
| Salinity | *+0.0008 ± 0.0020* | *1950-2018* | *-0.0019 ± 0.0025* | *1950-1994* | +0.0039 ± 0.0070 | 1995-2018 |
| Oxygen | *+0.028 ± 0.474* | *1950-2018* | *-0.183 ± 1.190* | *1950-1994* | *-0.675 ± 0.819* | *1995-2018* |
| Nitrate | +0.051 ± 0.088 | 1966-2018 | +0.257 ± 0.220 | 1966-1988 | *-0.011 ± 0.530* | *1995-2018* |
| Silicate | *-0.052 ± 0.077* | *1968-2018* | *+0.020 ± 0,139* | *1968-1994* | *-0.161 ± 0.444* | *1995-2018* |
| Phosphate | *+0.002 ± 0.005* | *1957-2018* | +0.011 ± 0.008 | 1957-1988 | *-0.001 ± 0.009* | *1995-2018* |
| | | | | | | |
| Area D | 5°S-5°N, 105°W-115°W, 50-300 m | | | | | |
| Temperature | *+0.003 ± 0.019* | *1955-2016* | *+0.076 ± 0.209* | *1955-1975* | *-0.004 ± 0.094* | *1979-2016* |
| Salinity | *-0.0000 ± 0.0018* | *1955-2016* | *-0.0017 ± 0.0068* | *1955-1975* | *+0.0001 ± 0.0022* | *1979-2016* |





| 306 | Oxygen | -0.643 ± 0.367 1957-2016 | *-2.390 ± 3.100 1957-1971* | -0.825 ± 0.825 1979-2016 |
| 307 | Nitrate | *+0.033 ± 0.166 1964-2016* | *+0.329 ± 14.90 1964-1968* | *+0.223 ± 0.272 1983-2016* |
| 308 | Silicate | *-0.001 ± 0.147 1967-2016* | +1.410 ± 0.921 1967-1970 | *+0.053 ± 0.546 1983-2016* |
| 309 | Phosphate | *-0.002 ± 0.013 1957-1994* | *+0.005 ± 0.046 1957-1971* | +0.035 ± 0.021 1983-1994 |
| 310 | | | | |
| 311 | Area E | 5°S-5°N, 165°W-175°W, 50-300 m | | |
| 312 | Temperature | *-0.006 ± 0.020 1950-2016* | +0.026 ± 0.060 1950-1976 | -0.010 ± 0.051 1977-2016 |
| 313 | Salinity | +0.0005 ± 0.0026 1950-2016 | +0.0005 ± 0.0100 1950-1979 | +0.0000 ± 0.0058 1977-2016 |
| 314 | Oxygen | -0.361 ± 0.224 1950-2016 | *-0.192 ± 0.781 1950-1975* | *-0.570 ± 0.574 1977-2016* |
| 315 | Nitrate | +0.054 ± 0.062 1961-2016 | *+0.159 ± 0.366 1961-1975* | *+0.105 ± 0.154 1977-2016* |
| 316 | Silicate | *-0.046 ± 0.148 1956-2016* | +0.172 ± NaN 1956-1975 | *+0.085 ± 0.174 1977-2016* |
| 317 | Phosphate | -0.003 ± 0.003 1950-2009 | *-0.002 ± 0.007 1950-1979* | *+0.005 ± 0.022 1990-2009* |
| 318 | | | | |
| 319 | Area F | 5°S-0°N, 90°E-98°E, 50-300 m | | |
| 320 | Temperature | *-0.002 ± 0.028 1960-2016* | *+0.004 ± 0.056 1960-1990* | *+0.033 ± 0.163 1995-2016* |
| 321 | Salinity | *+0.0020 ± 0.0025 1960-2016* | +0.0049 ± 0.0038 1960-1996 | *+0.0043 ± 0.0071 1995-2016* |
| 322 | Oxygen | *-0.221 ± 0.263 1960-2016* | *-0.098 ± 0.765 1960-1990* | *+0.123 ± 1.220 1995-2016* |
| 323 | Nitrate | *+0.036 ± 0.174 1962-2007* | *-0.130 ± 0.581 1962-1984* | -0.207 ± NaN 1995-2007 |
| 324 | Silicate | *+0.033 ± 0.410 1960-2007* | *+0.173 ± 0.619 1960-1990* | -0.368 ± NaN 1995-2007 |
| 325 | Phosphate | *+0.003 ± 0.009 1960-2007* | *+0.003 ± 0.014 1960-1989* | -0.015 ± NaN 1995-2007 |
| 326 | | | | |
| 327 | | | | |
| 328 | | | | |
| 329 | | | | |
| 330 | | | | |
| 331 | | | | |



**Figure 3:** Annual mean oxygen concentration for years available and trends for the layer 50 to 300 m in µmol kg$^{-1}$ plotted for the available years in the time period 1950 to 2018 (dashed red line) and for the positive and negative periods of the AMO in the Atlantic (a-c), the PDO in the Pacific (d,e) and the IOD in the Indian Ocean (f) as solid red lines. The AMO, PDO and IOD are shown as grey lines. The change of AMO status in 1963 and 1995, the change of the PDO phase in 1977, 1999 and 2013 and the IOD in 1990 are marked by dotted vertical lines. El Niño years defined as strong are marked by an additional magenta circle, strong La Niña years by an additional blue square.



**Figure 4:** Annual mean nitrate concentration for years available and trends for the layer 50 to 300 m in µmol kg⁻¹ plotted for the available years in the time period 1950 to 2018 (dashed red line) and for the positive and negative periods of the AMO in the Atlantic (a-c), the PDO in the Pacific (d,e) and the IOD in the Indian Ocean (f) as solid red lines. For area A the nitrate measurements in 1974 were removed as the 50-300 m mean was much too low 2.93 µmol kg⁻¹ and for area D the nitrate measurements were removed in 1970 which were too high (30.28 µmol kg⁻¹). The AMO, PDO and IOD are shown as grey lines. The change of AMO status in 1963 and 1995, the change of the PDO phase in 1977, 1999 and 2013 and the IOD in 1990 are marked by dotted vertical lines. El Niño years defined as strong are marked by an additional magenta circle, strong La Niña years by an additional blue square.

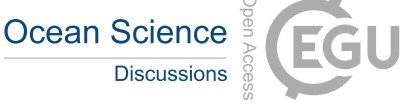

While oxygen decreased in all areas except for area C in the eastern tropical South Atlantic for the entire time period in the
50 to 300 m layer, nitrate increased in all areas (Figure 4). Phosphate also increased in the Atlantic and Indian Ocean areas,
while it decreased in the 2 areas of the equatorial Pacific Ocean (Table 2). Silicate decreased in the Atlantic and Pacific areas
but increased in the eastern Indian Ocean (area F). The temperature decreased in the central equatorial Pacific and the eastern
Indian Ocean (areas E and F) as is the case for these areas also in the 300 to 700 m layer. Surprisingly at the equatorial area
in the Atlantic (area B) the temperature in the 50 to 300 m layer decreased while it increased in the 300 to 700 m layer. The
50 to 300 m layer at the equator is governed by the eastward flowing Equatorial Undercurrent (EUC) while in the 300 to 700
m layer the westward flowing Intermediate Undercurrent (IUC) is located which might have an influence on the temperature
change over time.  The salinity in the 50 to 300 m layer increased in all areas except for a stagnant salinity concentration in
the eastern tropical Pacific Ocean (area D; Table 2).
The largest amount of years with available nutrient data exists in area A in the Atlantic Ocean. The long-term trends in area
A for temperature and oxygen for the 50 to 300 m layer (Table 2, Figure 5a,c) are similar as for the deeper layer 300 to 700
m (Table 1), however with increased variability near the surface most likely influenced by the seasonal cycle. For the 3
Atlantic areas A, B and C the long-term 50 to 300 m trend decreased for oxygen and silicate, and increased for salinity,
nitrate, phosphate and temperature, the latter except for temperature in area B with a weak not significant temperature
decrease. In the Atlantic, the equatorial station B shows higher mean 50 to 300 m layer temperature, salinity and oxygen and
lower mean nitrate, silicate and phosphate values compared to the off-equatorial stations A and C (Table 3) and shows the
eastward transport of oxygen-rich water with the EUC to the low oxygen regions in the eastern tropical Atlantic.


















**Figure 5:** Annual mean oxygen concentration for years available and trends for the layer 50 to 300 m in µmol kg⁻¹ plotted for the available years in the time period 1950 to 2018 (dashed red line) and for the positive and negative periods of the AMO in the Atlantic at area A for temperature (a) in °C, salinity (b), oxygen (c) in µmol kg⁻¹, nitrate (d) in µmol kg⁻¹, silicate (e) in µmol kg⁻¹ and phosphate (f) in µmol kg⁻¹. The AMO is shown as a grey line. The change of AMO status in 1963 and 1995 is marked by dotted vertical lines. El Niño years defined as strong are marked by an additional magenta circle, strong La Niña years by an additional blue square.







**Table 3.** Mean values for the layers 50-300 m and 300 to 700 m of temperature in °C, salinity and solutes in µmol kg$^{-1}$ in the
Atlantic Ocean (areas A, B, C), in the Pacific Ocean (areas D, E) and in the Indian Ocean (area F).

| Parameter | area A | area B | area C | area D | area E | area F |
|---|---|---|---|---|---|---|
| **50-300 m** | | | | | | |
| Temperature | 13.239 | 14.957 | 13.378 | 13.906 | 19.292 | 16.999 |
| Salinity | 35.350 | 35.432 | 35.333 | 34.888 | 35.116 | 35.010 |
| Oxygen | 91.045 | 109.888 | 55.178 | 70.059 | 141.116 | 96.157 |
| Nitrate | 22.518 | 17.754 | 27.438 | 23.629 | 13.771 | 18.755 |
| Silicate | 9.718 | 7.523 | 10.742 | 20.586 | 10.330 | 19.923 |
| Phosphate | 1.414 | 1.148 | 1.729 | 1.825 | 1.153 | 1.404 |
| | | | | | | |
| **300 – 700 m** | | | | | | |
| Temperature | 9.156 | 7.417 | 7.799 | 8.404 | 8.354 | 9.805 |
| Salinity | 35.020 | 34.673 | 34.726 | 34.656 | 34.646 | 34.966 |
| Oxygen | 62.194 | 104.494 | 47.251 | 32.991 | 72.027 | 67.922 |
| Nitrate | 34.457 | 31.280 | 39.284 | 35.662 | 32.967 | 31.437 |
| Silicate | 17.505 | 19.312 | 22.054 | 43.906 | 38.101 | 38.401 |
| Phosphate | 2.097 | 2.038 | 2.473 | 2.659 | 2.458 | 2.146 |



In the 50 to 300 m layer of area A despite the expected generally lower oxygen during positive AMO phase oxygen
increased in the positive AMO phase after 1995 (Figure 3a) different to the decrease in the 300 to 700 m layer (Figure 2a).
During the positive AMO phase after 1995 in the 50 to 300 m layer of area A trends in temperature, oxygen, nitrate, silicate
and phosphate (Figure 5) changed sign compared to the long-term trend while salinity showed for this period the same
continuous trend as the positive long-term trend. In contrast none of these parameters changed during positive AMO
compared to the long-term trend at the 50 to 300 m layer in the equatorial Atlantic in area B. In the tropical North Atlantic
(area A) and the equatorial Atlantic (area B) the La Niña events showed lower than normal oxygen concentrations especially
for the years 1973/74, 1975/76 and 2010/11 (Figure 3a,b). These years were not covered in the eastern tropical South
Atlantic (area C). In the equatorial area B, the El Niño years 1965/66, 1972/73, 1987/88 and 1991/92 showed slightly higher
than normal oxygen concentrations (Figure 3b). Although not true for all ENSO events, there seems to be some influence of
the La Niña and El Niño events in the eastern tropical and equatorial Atlantic, which might be due to the various types with
different hydrographic impact of ENSO events described in literature.





In eastern Pacific regions near the Galapagos Islands (2-5°S, 84-87°W) and near the American continent in the CalCofi
region (34-35°N, 121-122°W) and the Peru region (7-12°S, 78-83°W) oxygen increased and nutrient decreased in the 50 to
300 m layer during the negative PDO phase before 1977 with opposing trends during the positive PDO phase after 1977
(Stramma et al. 2020). Different to the eastern Pacific the eastern and central and equatorial areas D and E (Table 2) don't
show the reversed trends in oxygen and nutrients, however temperature and salinity indicate a reversal with the PDO phase
as the PDO index encapsulates the major mode of sea surface temperature variability in the Pacific. On a global scale the
long-term SST trend 1901-2012 was positive everywhere except for a region in the North Atlantic (IPCC 2013, Fig. 2.21).
For 1981 to 2012, while the western Pacific showed a warming trend, a large region with decreasing SST's was seen in the
eastern and equatorial Pacific Ocean (IPCC 2013, Fig. 2.22). This agrees to the temperature reversal seen in areas D and E.
However, if the time period after 1977 is looked at separately for the positive PDO phase 1977 to 1999 and the negative
PDO phase 1999 to 2013 as in the layer 300 to 700 m also the layer 50 to 300 m shows the expected high oxygen
concentrations in the period 1977 to 1990 and lower oxygen concentrations during 1999 to 2010 (Figure 3d,e).
Although ENSO is a signal originating in the Pacific the equatorial Pacific areas D and E show no obvious oxygen
concentration changes related to ENSO events (Figure 3d,e). The central equatorial Pacific area E shows the highest mean 50
to 300 m temperature and oxygen concentrations and the lowest nitrate concentrations of all six areas investigated (Table 3).
The low nitrate and phosphate and lower silicate compared to the eastern equatorial area D shows the nutrient concentration
decreasing westward in the equatorial Pacific in the 50 to 300 m layer (Stramma et al., 2020; their Figure 2). The principal
source of nutrients to surface water is vertical flux by diffusion and advection and by regeneration (Levitus et al., 1993). At
the sea surface airborne nutrient supply from land is contributed as well as terrestrial runoff of fertilizer-derived nutrients and
organic waste add nutrients to the ocean (Levin, 2018).  The tongue of high nutrient concentrations at the equatorial Pacific
compared to the subtropical Pacific results from upwelling near the American shelf (Levitus et al., 1993) and equatorial
upwelling.
In the eastern Indian Ocean as in the 300 to 700 m layer the temperature in the 50 to 300 m layer (Table 2) decreases and
indicates other processes related to the oxygen decrease instead of warming. In the Indian Ocean the IOD shows large
variability on shorter time scales. Observations indicate that positive IOD events prevent anoxia off the west coast of India
(Vallivattathillan et al. 2017). The IOD is very variable with a slightly higher index after 1990. The few oxygen
measurements in the 50 to 300 m layer indicate in area F until 1990 high mean oxygen concentrations with a decrease in
oxygen and after 1990 low oxygen concentrations with an increase in oxygen (Figure 3f). The higher oxygen concentrations
before 1990 and lower oxygen concentrations afterwards are also visible in the 300 to 700 m layer (Figure 2f). The ENSO
events don't indicate a visible influence on the oxygen concentration in area F. The four La Niña events between 1988 and
2008 were either below or above the mean trend-line, the same is true for the two El Niño events in 1973/74 and 1987/88
(Figure 3f).






## 4 Discussion and Summary

The time-series expansion of the six areas in the tropical oceans to the period 1950 to 2018 years showed a similar decrease in oxygen in the 300 to 700 m layer as described for the 1960 to 2008 period. Therefore, despite the overlying variability the long-tern deoxygenation in the tropical oceans is continuous for the 68-year period. This confirms the indicated importance on the 48-year period (Stramma et al. 2008) of the oxygen trend for future oceanic scenarios. The salinity trends are weak and not statistically significant, except for a salinity increase of $0.0012$ yr$^{-1}$ in the 300 to 700 m layer of area A in the tropical Northeast Atlantic. A consistent pattern in vertical sections in the Pacific Ocean is that nitrate and phosphate increase with depth to about 500 m, with a slight maximum at intermediate depths of 500–1500 m, while silicate continues to increase with depth (Fiedler and Talley, 2006) which is well visible in the higher mean concentrations in the 300 to 700 m layer in comparison to the 50 to 300 m layer (Table 3).

The temperature trends were positive in the three Atlantic areas, but positive or negative in relation to the time period included in the Pacific and Indian Ocean areas, hence we can conclude that the decreasing oxygen is not fully coupled to the local temperature change.  As the decline of oxygen in the tropical Pacific was not accompanied by a temperature increase, Ito et al. (2016) concluded that the cause of the oxygen decline must include changes in biological oxygen consumption and/or ocean circulation. Modelling the depth range 260 to 710 m depth range for 1990s-1970s the region of our areas D and E were mainly influenced by circulation variability (Ito et al., 2016).

Enhanced temperature differences between land and sea could intensify upwelling winds in eastern upwelling areas (Bakun, 1990). Observed and modelled changes in wind in the Atlantic and Pacific over the past 60 years appears to support the idea of increased upwelling winds (Sydeman et al., 2014). Coastal and equatorial upwelling enhance nutrients in the upper ocean, therefore the increase of nutrients in the eastern and equatorial oceans might be caused by winds intensifying upwelling. More nutrients in the surface layer enhances production and subsequently export and thus at greater depth its decay with increased respiration reduces the oxygen content. The sinking flux of organic matter, which over time depletes oxygen, while adding carbon and nutrients to subsurface waters, is known as the biological pump (Keeling et al., 2010) and could cause the often observed opposite trends in oxygen and nutrient trends in the 50 to 300 m layer investigated here. In the 50 to 300 m layer oxygen, temperature, salinity and nutrients showed long-term trends, which were different in the three ocean basins. Nitrate increased in all areas. Phosphate also increased in the Atlantic and Indian Ocean areas, while it decreased in the two areas of the equatorial Pacific Ocean. The phosphate increase in the Atlantic Ocean might be related to a continuous phosphate supply with the Saharan dust distributed over the Atlantic Ocean with the wind (Gross et al. 2015). Silicate decreased in the Atlantic and Pacific areas but increased in the eastern Indian Ocean. Often the expected inversely trend of oxygen and nutrients caused by remineralization of marine detritus (Whitney et al. 2013) was observed, however variations based on other drivers influence the nutrient trends.

An influence of ENSO years on the oxygen distribution with lower mean oxygen concentrations in the 50 to 300 m layer in La Niña years and larger oxygen concentrations in El Niño years was visible in the tropical North Atlantic and equatorial Atlantic. No clear impact of ENSO was observed in the tropical South Atlantic and the Pacific and Indian Ocean areas (C to





F).  To construct time series in areas with low data availability measurements from larger areas had to be taken into account.
As a result, there is a small possible bias due to the distribution of the measurements within the area and due to gaps in the
time line. In addition, there might be variations due to the measurement techniques for oxygen and nutrients and the use of
different reference material used for nutrient measurements or applied bias for nutrient measurements. Utilization of
historical nutrient data to assess decadal trends has been hindered by their inaccuracy, manifested as offsets in deep water
concentrations measured by different laboratories (Zhang et al., 2000). Although the trends are often not 95% significant the
results indicate existing trends and climate related changes, which might be verified with additional data in the future.
Although the data base is small especially for nutrients there is an indication that variability overlain on the long-term trends
is connected to climate modes as was found in the eastern Pacific with reversing trends related to the PDO (Stramma et al.,
2020). The six areas of the tropical ocean basins indicate some connection to the climate modes of the 3 ocean basins. In the
tropical eastern North Atlantic (area A) there is some dependence with the AMO. In the equatorial Pacific areas D and E a
connection to the PDO is visible when the positive PDO phase 1977 to 1999 and the negative PDO phase 1999 to 2013 are
looked at separately. In the eastern tropical Indian Ocean there seems to be some dependence to the state of the IOD, despite
the fact that the IOD varies more on shorter time scales and the IOD change in 1990 is weak.
Future measurements of temperature, salinity, oxygen and nutrients could lead to more stable results determining trends and
their variability to better understand the influence of climate change on the ocean ecosystem and prepare future predictions
of ocean oxygen from Earth System Models (Frölicher et al., 2016). Making existing nutrient data public which are so far
not in public data bases and modelling efforts on oxygen and nutrient changes would further improve the understanding of
oxygen and nutrient variability and its biological influence e.g. on fishery. First ecosystem changes like habitat compression
can be observed and negative impacts are expected on biological regulation, nutrient cycling and fertility, and sea food
availability with an increasing risk of fundamental and irreversible ecological transformations (Hoegh-Guldberg and Bruno,

522    2010).




*Data availability.* The AMO time series was taken from https://www.esrl.noaa.gov/psd/data/timeseries/AMO/ (ESRL,
Climate    time    series,    status    17.02.2020).    The    Indian    Ocean    Dipole    Mode    was    taken    from
https://www.esrl.noaa.gov/psd/gcos_wgsp/Timeseries/Data/dmi.long.data on 3 March 2020. The yearly PDO data were
taken from http://ds.data.jma.go.jp/tcc/tcc/products/elnino/decadal/annpdo.txt on 9 July 2020 from the Japan Meteorological
Society covering the period 1901 to 2019.
The bottle data from cruises in 2016 at 170°W (096U2016426_hyd1.csv) and at 110°W (33RO20161119_hyd1.csv) were
downloaded from the CCHDO at the University of California San Diego (https://cchdo.ucsd.edu, CCHDO, 2020) on 8
November 2018.



The added ship cruises are contained for CTD data in the data sets of https://doi.org/10.1594/PANGAEA for RV Meteor
cruises M120, https://doi.pangaea.de/10.1594/PANGAEA.868654 (Kopte and Dengler 2016), M130
https://doi.pangaea.de/10.1594/PANGAEA.903913 (Burmeister et al. 2019), M131
https://doi.pangaea.de/10.1594/PANGAEA.910994 (Brandt et al. 2020), M145
https://doi.pangaea.de/10.1594/PANGAEA.904382 (Brandt and Krahmann, 2019), and, for RV Meteor cruise M148
https://doi.pangaea.de/10.1594/PANGAEA.????, and RV Merian 07 https://doi.pangaea.de/10.1594/PANGAEA.???? and
for nutrient data in the data sets of Merian MSM10/1 https://doi.pangaea.de/10.1594/PANGAEA.775074 (Tanhua et al.
2012), RV Poseidon 250 https://doi.pangaea.de/10.1594/PANGAEA,????, M68/2
https://www.ncei.noaa.gov/data/oceans/ncei/ocads/data/0108078/, M83/1
https://doi.pangaea.de/10.1594/PANGAEA.821729 (Tanhua 2013). M97
https://doi.pangaea.de/10.1594/PANGAEA.863119 (Tanhua 2016), Meteor M106
https://doi.pangaea.de/10.1594/PANGAEA.????, Meteor M119 https://doi.pangaea.de/10.1594/PANGAEA.????, M120
https://doi.pangaea.de/10.1594/PANGAEA.????, Meteor M130 https://doi.pangaea.de/10.1594/PANGAEA.913986 (Tanhua
2020), Meteor M131 https://doi.pangaea.de/10.1594/PANGAEA.????, Meteor M145
https://doi.pangaea.de/10.1594/PANGAEA.????, and Meteor M148 https://doi.pangaea.de/10.1594/PANGAEA.????. Open
references (shown as ????) will be made available before publication.

*Author contributions.* L. Stramma conceived the study and wrote the manuscript. S. Schmidtko compiled the data for the
time series, collected further references and discussed and modified the manuscript.

*Competing interests.* The authors declare that they have no conflict of interest.

555 .

*Acknowledgements.* Financial support was received through GEOMAR and the Deutsche Forschungsgemeinschaft (DFG) as
part of the "Sonderforschungsbereich 754: Climate-Biogeochemistry Interactions in the Tropical Ocean".

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
