# Peer review of "Oxygen and nutrient trends in the Tropical Oceans"

_Ocean Science, 2020_

## Referee Comment (RC1) · Anonymous Referee #1 · 29 Jan 2021

This is a very interesting and potentially significant paper for documenting important long term environmental changes in the ocean. The authors have amplified and extended the time coverage of their previous work with data from the 1950's to the present day and with additional parameters and wider geographic coverage. This paper is especially interesting for demonstrating the association of trends with the various climate indices pertinent for different oceanic regions. Sub-dividing the climate indices into their modes and separating the environmental trends by mode (in addition to the overall trend) was a novel approach that illuminated some of the complexities of tracking and interpreting these changes. This will be an important contribution to our understanding of global climate change effects in the ocean. Specific comments and suggestions are below.

Abstract: The abstract begins with a reference to the "vertical expansion of the intermediate-depth low-oxygen zones", a topic developed in the earlier Stramma et al. (2008) paper. The present manuscript does not really discuss "vertical expansion" and I suggest modifying this introductory abstract sentence so that it focuses on the present paper.

Methods: I am not a specialist with the statistical methodology (a limitation discussed in advance by email with the editor). However, many readers of this paper are likely to also be non-specialists in this regard. In order to make this paper more comprehensible to the broader global change community, a simple non-technical summary paragraph of how the data were chosen and assembled to get a single point representing 1 year per depth zone and area, and the limitations of the dataset and methodology, would be valuable and would prevent mis-interpretation. Although I read the referenced methods in Schmidtko et al. (2017), I still have questions (and most general readers would probably not refer back to that reference).

Some methodological questions that could be addressed in a summary paragraph include:

Were only individual bottle sample data used or were data from continuous electronic sensors from the last few decades included (vertical profiles from the many oceanographic instruments that include CTD sensors)? If electronic profiles were used, how were they condensed into 1 value for that depth interval? Were any data from Argo floats used since temporal and geographic coverage is expanding (although extended geographical oxygen coverage is only just beginning with the Biogeochemical Argo floats)? If these various electronic profilers were not used, a major source of modern data was ignored, and this limitation should be acknowledged. The discussion could include suggestions about how these continuous profiles might be applied to future trend calculations.

How many individual data points (or specific CTD casts) were included for each year?

This is especially important for interpreting the earliest records. Could this be included in a supplementary table?

The authors do discuss how they accounted for chemical measurement improvements over this time period. However, since the 1950's, there have also been substantial improvements in location (satellite-based) and depth (electronic sensor) data. Was there any consideration for those sampling changes and uncertainty of those parameters in earlier data? That should at least be mentioned.

The authors mention standard vertical depth levels (line 126). What are the depths and how were they used? As someone who deals with the small-scale vertical and temporal variability of oxygen through some of these depth strata, I remain confused about how that single value was derived, what it represents, and how robust that calculated value is. Is this supposed to be an indicator of the concentration of oxygen in the OMZ? Small changes in depth (and location and time) within their indicated depth zones can have quite different oxygen concentrations, as we now know from continuous profiling sensors and short-term replicated profiles. Since many of the long-term trends are slight, small changes in this value (for example from bottles offset by < 10 m or taken a few hrs apart) could have large consequences for the calculated trends. In the discussion, the authors should include more about the uncertainty associated with these issues, how to account for that, and how to improve future predictions.

Results and Figures

Tables 1 and 2. It would be helpful to have more space between items or some other font indicator (bold?). It took several readings to notice the year listing since this blended in with the other numbers.

Fig. 2 and 3. The plots should be labelled A, B, C etc. There should be bigger tick marks for the decadal divisions on the x axis so those are visually differentiated from the climate index modal divisions. The caption should include a sentence noting that the X's in the plots represent an annual value (one X per year). There should also be

a sentence to point out that the Y axes for oxygen (and nitrate in Fig. 4) differ in each plot.

Fig 5. First line of the caption should delete "oxygen" and its units since this figure shows different variables in each graph.

It would be very interesting to see all the overall trend lines for a particular variable (oxygen in the two depth zones, nitrate) plotted on a single graph (one graph for each variable) with a single Y axis range. This would allow a visual comparison of the geographic variability in the strength of these trend lines. This could be a supplemental figure.

Discussion This paper provides strong support for the trend of increasing deoxygenation worldwide at mid ocean depths. Many interesting possibilities are discussed to explain the trends in the different ocean areas and the interactions of different variables with regional physical oceanography, the climate modes, and each other. However, this also becomes confusing with details. It would be helpful to have some summary conclusions, perhaps a numbered list by geographic area. I would also like to see more discussion about the broader implications of these trends especially for ocean biology and human impacts. The authors could also bring in some mention of how their results could contribute in the future to some of the major climate change discussions and documents (IPCC report, UN Ocean Decade, etc.)

Minor wording comments: Line 47: sinking, transport, and subduction into the deep ocean (the biological pump includes more processes than sinking) Line 73: extent Line 74: biologically Line 97: despite the fact that the low Line 434: nutrients Line 441: agrees with Line 443: sentence? Line 452: adding Line 463: trend-line; Line 478: areas. Hence, Line 484: appear Line 485: ocean; therefore, Line 495: inverse Line 496: observed; however, Line 519: fisheries

---

## Referee Comment (RC2) · Anonymous Referee #2 · 3 Mar 2021

The manuscript describes changes in dissolved oxygen and nutrients which might have various and important impacts on the ocean ecosystem. As the authors note, the manuscript confirms conclusions done in the previous work by Stramma et al (2008), which describes trends in the same selected geographical areas.

1) My main concerns regarding the significance of the presented results relates to the poor observational basis. In the concluding remarks the authors themselves acknowledge this fact and point to the necessity for further verification in the future.

The visual inspection of the calculated trends shows high variability of the yearly parameter concentration values imposed on a much weaker climatological signal. Some of the trends are not significant at 95 percent level. The others, even formally significant, leave the impression that the removal of just few data points would lead to significantly different trend estimates. Here, increasing the number of investigated areas of similar size , or increasing the size of the investigated areas could help to confirm (or not to confirm!) the robustness of the presented results.

2) Using optimally averaged yearly values which in turn are used to estimate trends is justified, as the averaging procedure acts a a smoother. However, I would appreciate the elaboration about the possible errors related to the mapping scheme, and, because of the data paucity, even these averaged data might be linked to a relatively large errors.

3) The authors do provide measurement precisions for the modern measurements and offset estimates for the older data. The reference is done to the paper by Tahua et al., 2010.

Even for temperature which is easier to measure compared to other parameters, systematic instrumental errors pose a big problem in estimating the ocean heat content changes. Possible instrumental biases in oxygen and nutrient data is even a bigger issue, which was treated, for instance, during the WOCE time. Here, references to Johnson et al., 2001, and to Gouretski and Jancke, 2001 could be added to the reference list. From the manuscript text it is not clear, whether the original data were corrected for systematic biases, or not. More discussion of possible bias impact is needed.

4) Several areas show a certain shift in oxygen concentration for the data after 2000 (Fig.2a,b,d,f; Fig.3 a,e) Could these change in concentration be due to unexplored biases?

5) it is interesting to know, how sensitive the calculated trends are to the thickness of the layers (the fixed thickness of 250 and 400 meters are used for the presented results).

6) Can the yearly parameter values be seasonally biased considering the poorness of the data basis??

Minor comments:

1) I do not see much use in presenting mean parameter values for the investigated areas without providing the number of available original profiles and standard deviations (Table 3) 2) Please, indicate in the figure captions, that crosses denote the annual parameter values used to calculate trends

Line 13: change indicates to indicate

Lines 125-126: probably the word "interpolated" is missing after the word "profiles" (Line 126)

Line 267: change nutrient to nutrients

---

## Author Comment (AC1) · 21 Apr 2021

Our replies are added to the reviewer's comment. A more structured pdf-version is added as supplement.

We thank both reviewers for their comments, which helped to improve the revised manuscript.

Reviewer(s)' Comments to Author:

Reviewer: 1

Abstract: The abstract begins with a reference to the "vertical expansion of the intermediate-depth low-oxygen zones", a topic developed in the earlier Stramma et

al. (2008) paper. The present manuscript does not really discuss "vertical expansion" and I suggest modifying this introductory abstract sentence so that it focuses on the present paper.

Thanks, good remark. The first sentence focusses now on the oxygen decrease instead of the vertical expansion.

Methods: I am not a specialist with the statistical methodology (a limitation discussed in advance by email with the editor). However, many readers of this paper are likely to also be non-specialists in this regard. In order to make this paper more comprehensible to the broader global change community, a simple non-technical summary paragraph of how the data were chosen and assembled to get a single point representing 1 year per depth zone and area, and the limitations of the dataset and methodology, would be valuable and would prevent mis-interpretation. Although I read the referenced methods in Schmidtko et al. (2017), I still have questions (and most general readers would probably not refer back to that reference).

We added a more non-technical summary paragraph in the Data and methods chapter on page 5. With regard to limitation of the method also the text in the Discussion and Summary chapter was modified and more details added that are easier to comprehend the steps taken in this manuscript and in the references. See also our answer below.

Some methodological questions that could be addressed in a summary paragraph include: Were only individual bottle sample data used or were data from continuous electronic sensors from the last few decades included (vertical profiles from the many oceanographic instruments that include CTD sensors)? If electronic profiles were used, how were they condensed into 1 value for that depth interval? Were any data from Argo floats used since temporal and geographic coverage is expanding (although extended geographical oxygen coverage is only just beginning with the Biogeochemical Argo floats)? If these various electronic profilers were not used, a major source of modern data was ignored, and this limitation should be acknowledged. The discussion could

include suggestions about how these continuous profiles might be applied to future trend calculations.

In the additional paragraph in the Data and methods section it is now described that bottle and CTD data are used and that float data were not included as measurements from our own floats showed drifts probably due to biological activity at the sensors which would lead to erroneous trends. The possibility to use float data in future once drifts can be removed is mentioned in the Discussion and Summery section. The main focus of this manuscript is to compare the extended time period with the Stramma et al. (2008) results, this paper is even today highly cited. Therefore, we used the similar methods to make the results comparable (mentioned now in the additional paragraph in the data and Methods chapter). CTD data are available in 1 dbar steps. We used only the CTD values 5 dbar apart to reduce the amount of data, as also the vertical gridding was made on 5 dbar steps. This information is now included in the supplement text. Using an objective mapping scheme described in the manuscript the available data were mapped on 5 dbar steps and then the mean value and standard deviation for the two depth intervals 50 to 300 m and 300 to 700 m was computed. This in now explained in the text.

How many individual data points (or specific CTD casts) were included for each year? This is especially important for interpreting the earliest records. Could this be included in a supplementary table?

The data points and CTD profiles vary for each year in each area. A table with the numbers of data points would not show how large the bias of the computed mean oxygen value is. However, we mention now the possible influence of the fewer data point within a selected depth layer in the additional paragraph in "Data and methods".

The authors do discuss how they accounted for chemical measurement improvements over this time period. However, since the 1950's, there have also been substantial improvements in location (satellite-based) and depth (electronic sensor) data. Was there

any consideration for those sampling changes and uncertainty of those parameters in earlier data? That should at least be mentioned.

We mention now in the Data and methods paragraph the uncertainty caused by less accurate depth measurements from bottle data with CTD-depth measurements. As all measurements within each area independent from the geographical location were used, the better satellite derived location should not influence the results presented here.

The authors mention standard vertical depth levels (line 126). What are the depths and how were they used? As someone who deals with the small-scale vertical and temporal variability of oxygen through some of these depth strata, I remain confused about how that single value was derived, what it represents, and how robust that calculated value is. Is this supposed to be an indicator of the concentration of oxygen in the OMZ? Small changes in depth (and location and time) within their indicated depth zones can have quite different oxygen concentrations, as we now know from continuous profiling sensors and short-term replicated profiles. Since many of the long-term trends are slight, small changes in this value (for example from bottles offset by < 10 m or taken a few hrs apart) could have large consequences for the calculated trends. In the discussion, the authors should include more about the uncertainty associated with these issues, how to account for that, and how to improve future predictions.

In the revised version more information is presented with regard to uncertainties. The reviewer is correct that time of measurements and vertical resolution play a role for uncertainties of the trends. Never-the-less on the larger scale, due to internal waves, depth uncertainties and other factors we can assume these errors to be noise the no systematic bias over time. Thus, any trends derived will be less certain though not biased in decline or increase of the parameter analyzed. Furthermore as mentioned above we use the same methods as in Stramma et al. 2008 to make the results comparable. With CTD oxygen measurements on 1 dbar steps the uncertainties between years will be much less than in comparison to bottle oxygen measurements in earlier
years. Nutrient trend computations could be improved if future nutrient measurements are made on defined standard depth levels to make the results better comparable. This is now explained in the Data and method section as well as in the supplementary file.

Results and Figures Tables 1 and 2. It would be helpful to have more space between items or some other font indicator (bold?). It took several readings to notice the year listing since this blended in with the other numbers.

The years in Tables 1 and 2 are now bold, hence it should be well recognizable what are the trends and what are the time periods.

Fig. 2 and 3. The plots should be labelled A, B, C etc. There should be bigger tick marks for the decadal divisions on the x axis so those are visually differentiated from the climate index modal divisions. The caption should include a sentence noting that the X's in the plots represent an annual value (one X per year). There should also be a sentence to point out that the Y axes for oxygen (and nitrate in Fig. 4) differ in each plot.

In Ocean Science subsets of figures are labelled a, b, c (not A, B, C)

Bigger tick marks now show the decadal divisions on the x-axis.

The (x) is included in the figure legends and a sentence added to point out that the y-axes changes.

Fig 5. First line of the caption should delete "oxygen" and its units since this figure shows different variables in each graph.

Thanks, "oxygen" was replaced by "parameter" and the units were deleted in the first line of Fig. 5 caption.

It would be very interesting to see all the overall trend lines for a particular variable (oxygen in the two depth zones, nitrate) plotted on a single graph (one graph for each variable) with a single Y axis range. This would allow a visual comparison of the geographic variability in the strength of these trend lines. This could be a supplemental figure.

As supplementary figures the trends for both oxygen layers and the nitrate trends are presented.

Discussion This paper provides strong support for the trend of increasing deoxygenation worldwide at mid ocean depths. Many interesting possibilities are discussed to explain the trends in the different ocean areas and the interactions of different variables with regional physical oceanography, the climate modes, and each other. However, this also becomes confusing with details. It would be helpful to have some summary conclusions, perhaps a numbered list by geographic area. I would also like to see more discussion about the broader implications of these trends especially for ocean biology and human impacts. The authors could also bring in some mention of how their results could contribute in the future to some of the major climate change discussions and documents (IPCC report, UN Ocean Decade, etc.)

We included near the end of the manuscript a paragraph listing the observations by ocean basins. The implication of oxygen trends for biology and human impacts quite large and a lot of literature exist. All aspects of oxygen trends are discussed in the different chapters of the IUCN report (2019) and this is now mentioned and referenced at the end of the summary.

Minor wording comments: Line 47: sinking, transport, and subduction into the deep ocean (the biological pump includes more processes than sinking) Line 73: extent Line 74: biologically Line 97: despite the fact that the low Line 434: nutrients Line 441: agrees with Line 443: sentence? Line 452: adding Line 463: trend-line; Line 478: areas. Hence, Line 484: appear Line 485: ocean; therefore, Line 495: inverse Line 496: observed; however, Line 519: fisheries

Thanks for pointing to the correct writing, all proposed changes were done in the modified manuscript.

Please also note the supplement to this comment:
https://os.copernicus.org/preprints/os-2020-123/os-2020-123-AC1-supplement.pdf

---

## Author Comment (AC2) · 21 Apr 2021

Our replies are added to the reviewers comments. A more structured pdf-version is added as supplement.

We thank both reviewers for their comments, which helped to improve the revised manuscript.

Reviewer(s)' Comments to Author:

Reviewer: 2

The manuscript describes changes in dissolved oxygen and nutrients which might have various and important impacts on the ocean ecosystem. As the authors note, the

manuscript confirms conclusions done in the previous work by Stramma et al (2008), which describes trends in the same selected geographical areas. 1) My main concerns regarding the significance of the presented results relates to the poor observational basis. In the concluding remarks the authors themselves acknowledge this fact and point to the necessity for further verification in the future. The visual inspection of the calculated trends shows high variability of the yearly parameter concentration values imposed on a much weaker climatological signal. Some of the trends are not significant at 95 percent level. The others, even formally significant, leave the impression that the removal of just few data points would lead to significantly different trend estimates. Here, increasing the number of investigated areas of similar size, or increasing the size of the investigated areas could help to confirm (or not to confirm!) the robustness of the presented results.

The main focus of this manuscript is to compare the extended time period with the Stramma et al. (2008) results, this paper is even today highly cited. Therefore, we used similar methods to make the results fully comparable (mentioned now in the additional paragraph in the data and Methods chapter). The data base for the manuscript was similar low as for the investigation of Stramma et al. (2008). Hence there is in this manuscript and in the earlier paper a larger uncertainty possible. For the 2008 paper later measurements in literature confirmed the decreasing oxygen trends. One focus of this manuscript is to compare the trends for the longer time period with the shorter one in the 2008-paper. For additional areas we would need an additional area with measurements covering a large part of the 1950 to present period, we are not aware of, as in the tropics there are very limited areas with a larger data base. Due to changing oxygen on the geographical locations an extension of the areas would include more uncertainties. Hence, we stayed with the same areas as in Stramma et al. (2008), but we describe these limitations more specific in the Chapters Data and methods, Discussion and Summary and in a supplementary file.

2) Using optimally averaged yearly values which in turn are used to estimate trends is

justified, as the averaging procedure acts as a smoother. However, I would appreciate the elaboration about the possible errors related to the mapping scheme, and, because of the data paucity, even these averaged data might be linked to relatively large errors.

The reviewer is correct, this procedure smoothes the dataset to a degree. We added more information to the supplemental methods discussing the possible errors. In general, this mapping scheme is quite conservative and is more likely to show no trend than a trend for two reasons. First with a non-smoothed data set an oversampled anomalous year has significant impact on the whole timeseries, second with lower overall smoothed data points the uncertainty of a trend analysis is larger. With those points in mind, one can assume that the results, if statistically significant are a robust find.

3) The authors do provide measurement precisions for the modern measurements and offset estimates for the older data. The reference is done to the paper by Tahua et al., 2010. Even for temperature which is easier to measure compared to other parameters, systematic instrumental errors pose a big problem in estimating the ocean heat content changes. Possible instrumental biases in oxygen and nutrient data is even a bigger issue, which was treated, for instance, during the WOCE time. Here, references to Johnson et al., 2001, and to Gouretski and Jancke, 2001 could be added to the reference list. From the manuscript text it is not clear, whether the original data were corrected for systematic biases, or not. More discussion of possible bias impact is needed.

The offsets derived from an inter-cruise comparison described by Gouretski and Jancke (2001) is now included as well as the initial standard deviations of cross-over differences from Johnson et al. 2001. These papers are now used to describe a possible bias from the measurements. For heat content the systematic bias is one directional and addressed mainly expandable thermographs, such data is not used in this analysis. In the case of our data, we have evidence as referenced that the bias is more likely to be noise than systematic, thus not impacting any trend analysis.

4) Several areas show a certain shift in oxygen concentration for the data after 2000 (Fig.2a,b,d,f; Fig.3 a,e) Could these change in concentration be due to unexplored biases?

We think that these changes in oxygen concentration are related to the climate signals changing in all three oceans shortly before the year 2000. However, we included a discussion related to these changes in the supplementary text.

5) it is interesting to know, how sensitive the calculated trends are to the thickness of the layers (the fixed thickness of 250 and 400 meters are used for the presented results).

As mentioned above, one focus of this manuscript is a comparison with the Stramma et al. (2008) paper, therefore the layer 300 to 700 m was selected again. Depending on the parameter gradients at the boundaries of the layer, the trends will be different. However, as the oxygen trends for the 50 to 300 m layer and the 300 to 700 m are all negative (except for the 50 to 300 m layer of area C due to a local effect) the result of oxygen decrease is not related to the depth layer chosen. This is now described in the Discussion and Summary chapter.

6) Can the yearly parameter values be seasonally biased considering the poorness of the data basis??

As the annual data were computed independent of the season due to the poorness of the data basis, a seasonal influence might be possible. However, as the seasonal cycle in the tropics is weaker than in most subtropical and subpolar regions (Louanchi and Najjar, 2000) we mention it near the end of the Introduction and now also in the supplementary text.

Minor comments: 1) I do not see much use in presenting mean parameter values for the investigated areas without providing the number of available original profiles and standard deviations (Table 3)

The mean parameter values for a depth layer and year are used to compute the mean parameter values for the time period covered and here the number of years and the mean standard deviation are presented. The standard deviation is often large, as it is related to the variability of the annual mean parameter value and the strength of the trend during the measurement period, which is now explained in the text. As for each year a different amount of data is available, computing the standard deviation for each year would provide only information with regard to variability within a year, but not for the mean values presented in table 3. The fact that the mean parameter is derived from the annual values is now mentioned in Table 3.

2) Please, indicate in the figure captions, that crosses denote the annual parameter values used to calculate trends

Thanks, it is now mentioned that the annual parameter values are used to calculate trends.

Line 13: change indicates to indicate

As proposed changed to 'indicate'

Lines 125-126: probably the word "interpolated" is missing after the word "profiles" (Line 126)

Thank you, word "interpolated" added

Line 267: change nutrient to nutrients

As proposed changed to 'nutrients'

Please also note the supplement to this comment:
https://os.copernicus.org/preprints/os-2020-123/os-2020-123-AC2-supplement.pdf

---

## Referee Report (RR1)

The authors have addressed the concerns of the first review in a substantive way. I especially appreciate the new figure S1 and the addition of pertinent details on the methods. There are still a few minor copy editing and English language issues (especially in the new Supplement text).

---

## Author Response (AR3)

**We thank both reviewers for their comments, which helped to improve the revised manuscript.**

Reviewer(s)' Comments to Author:

Reviewer: 1

For final publication, the manuscript should be
accepted subject to **technical corrections**

Suggestions for revision or reasons for rejection (will be published if the paper is accepted for final publication)
Excellent paper. There are still a few minor copy editing and English language issues, especially in the new Supplement text, which should be fixed before publication.

Referee Report:

The authors have addressed the concerns of the first review in a substantive way. I especially appreciate the new figure S1 and the addition of pertinent details on the methods. There are still a few minor copy editing and English language issues (especially in the new Supplement text).

**Thank you for accepting our revision of the first version of the manuscript. Please note, as reviewer 3 mentioned that the title of the manuscript did not fit well the focus on the 6 tropical regions investigated in a 2008 paper, we changed the title to better fit to the text of the manuscript. We read the text carefully and hopefully improved the text and supplemental readability.**

Reviewer(s)' Comments to Author:

Reviewer: 3

Review of the manuscript, entitled "Oxygen and nutrient trends in the Tropical Oceans" authored by Stramma and Schmidtko.

In this manuscript, the authors have intended to describe trends of oxygen and nutrients using long time series spanning from 1950 to 2018. This work is an extension of the authors' previous paper published in 2008 by including a more recent dataset and incorporating more variables into the analysis.
The manuscript presents an extensive set of dataset and therefore, contribute in enhancing our existing understanding of the said processes. However, as noted in the earlier comments by other reviewers, I have few major concerns and they are listed below:
1. The title of the manuscript says "trend of the Tropical Oceans". But actually, the analysis is limited to very few locations in the tropics. This is particularly true for the Indian Ocean. I

can understand that data limitation is the major problem to extend this study to a broader region, but then the title is grossly misleading.

**We changed the title to "Tropical deoxygenation sites revisited to investigate oxygen and nutrient trends", which clearly relates the new manuscript to the areas investigated in 2008.**

2. Another major problem with the analysis is that the calculated trends for oxygen and nutrients for most regions are either not statistically significant or the error associated with the trend is more than the trend itself. This indicates that goodness of fit for this trend calculations are very poor (can be seen easily from the plots as well), which means the spread of the data point is too large for a sensible linear fit. This casts a doubt on the reliability of these trends. However, there seems to be a definite decreasing (increasing) trend for the oxygen (nutrients) in the 300-700 layer, but the calculated trends may not be reliable.

**To make the readers aware of the unreliable trends based on the low data base we mention this already in the abstract as well as in the text and in the supplemental material. Nevertheless as it is pointed out, the poor data coverage do not allow a statistical proof of these trends, but the data distributions points to a possible change that is worth discussing.**

3. Authors have interpolated data into standard vertical depths and then averaged over 50-300 m and 300-700 m to calculate the trends for the respective variables.
(a) What are these standard depths?

**Most data is observed close to standard depths with variations. The approach here is using a 5dbar interpolation as now added in more details to the methods.**

(b) I believe, the depth ranges are chosen as an analogous to the ocean heat content approach. However, unlike temperature data, oxygen and nutrient data are much more sparse and therefore, may pose a problem while averaging. This is particularly problematic for the upper layer as some part of it is above the thermocline and some part within the thermocline and therefore, experience a strong gradient in oxygen concentration. Now, it may possible that in some years data above the thermocline is available and for other years within the thermocline. This will result in scatter concentration in time series similar to what we see here. Moreover, as the thermocline itself show a large shift due to ocean warming and various climate modes, the averaging across the depths (in case of data coverage is not uniform across the depth and year) will itself lead to a trend. How authors have addressed this issue is not very clear.

**Thank you for pointing this out again, we now explicitly state in the text that we start below the first nutrient/oxycline at 50m. And also mention now the possible influence of the thermocline shift and a possible influence on the trend computations for the upper layer more explicitly.**

(c) For vertical interpolation, a uniform Gaussian weighting method is applied with a uniform cut-off range in space and time across all the boxes and (top and intermediate) layers. Since the variability in the top layer is very different from the bottom layer, different weighting mechanisms may suit better for the different layers.

**As we interpolate the data in both depth layers on 5 dbar steps and the depth layers connect at 300 m depth, we think it would be good to keep the same weighting mechanisms to increase comparability.  For computing a nutrient or oxygen budget this would definitely be appropriate, but since the trend analysis uses the same distribution over time we are confident that we would not gain more reliable trends with varying mapping schemes.**

(d) In order to better appreciate the data inventory and the trends, is it possible to show number of data points available for each of the standard depths for each box? Maybe a plot with y-axis as depths and x-axis as years with the number of data points as colour will serve the purpose.

**The CTD data were reduced to 5 dbar steps and all data from one year within a 5 dbar interval were combined to one value in this 5 dbar interval. Due to different data sets from bottle or CTD data such a plot of the original data would be very variable for different years and could lead to confusion, but could not help to lead to better trend computations.**

4. Tables showing the trends are just too many details and may not be very useful to most readers. Is it possible to convert them as barplot with error bar for each variable? This will make life easy for the readers. The detailed tables can be provided as supplementary information for interested readers.

**We think in Table 1 and 2 the trend and 95% confidence interval should be presented to show which trends are not within the 95% confidence interval. The details on the number of profiles and the standard deviation were added to the table based on the request of one reviewer to add these values to the table.**

Overall, the trend calculations are not quite robust and also doubtful. There are a large number of caveats present in the data analysis. Therefore, I believe the title and abstract are exaggerated in nature. Authors need to spell these caveats upfront in the abstract so that the readers are not misled while interpreting these results.

**Yes, we write now in the abstract:   "Due to the low amount of data available the results are often not in the 95% confidence interval, but nevertheless indicate existing trends" and we changed the title of the manuscript, that the reader much better know that the results focus on few tropical areas and that the results have to be regarded with care.**